# Evaluating Fishing Capacity Based on DEA and Regression Analysis of China's Offshore Fishery

**Shuang Liu [1], Jia-Xin Sun [1], Chao Lyu [1,*], Ta-Jen Chu [2,3] and He-Xu Zhang [1]**

[1] College of Engineering Science and Technology, Shanghai Ocean University, Shanghai 201306, China; s-liu@shou.edu.cn (S.L.); dblvchao@163.com (J.-X.S.); zhanghexu2021@163.com (H.-X.Z.)
[2] Fisheries College, Jimei University, Xiamen 361021, China; chutajen@gmail.com
[3] Fujian Provincial Key Laboratory of Marine Fishery Resources and Eco-Environment, Xiamen 361021, China
\* Correspondence: clv@shou.edu.cn

**Abstract:** The analysis of offshore fishing capacity is of great significance and practical value to the sustainable utilization and conservation of marine fishery resources. Based on the 2004–2020 *China Fishery Statistical Yearbook*, data envelopment analysis (DEA) was applied for measuring fishing capacity using a number of fishing vessels, total power, total tonnage, and the number of professional fishermen as the input measures and the annual catch as the output measure. Capacity utilization had a calculated range from 80.7 to 100%, and its average is 93.5%. In the first four years of 2003–2007, the excess investment rate of fishing vessels, total tonnage, total power, and fishermen was low (<5%). There was a consistent sharp upward trend in 2007, a gradual downward trend from 2007 to 2015, and an upward trend after reaching a low point in 2015, with the highest gross tonnage of fishing vessels reaching 25.5%. Four regression models that incorporate machine learning algorithms are used, including Lasso, Ridge, KNN, and Polynomial Features. The goodness of fit for the four models was used as the evaluation index, and the offshore annual catch based on the evaluation index was proposed. The forecasting annual catch of the polynomial model can reach 0.98. Furthermore, a comparative simulation of the DEA incorporating the polynomial model was carried out. The results show that DEA can evaluate input factors under the conditions of a given range, and the polynomial model has more advantages in forecasting annual catches. Furthermore, the combined application of DEA and polynomial model was used to analyze and discuss the management policies of China's offshore fishery, which can provide help and reference for future management.

**Keywords:** fisheries; China offshore; fishing capacity; data envelopment method; machine learning algorithm

## 1. Introduction

In 2018, the total global capture of fisheries reached the highest level on record, reaching 96.4 million tons, which is an increase of 5.4% over the average level of the previous three years. China is the world's largest fishing nations in terms of its fishing fleet, the number of employees in the fishing industry, and marine capture production. Its annual marine catch in 2018 accounted for 15% of the world's total production [1]. However, roughly 57% of marine fish stock is overexploited or collapsed in China, and the rapid development of coastal cities has placed tremendous pressure on marine ecosystems [2]. Protecting offshore fishery resources, reducing fishing intensity, and strengthening fishing capacity are the core requirements for the sustainable development of marine fisheries. Overcapacity is a key factor contributing to decline in many of the world's fisheries. The FAO International Plan of Action for the Management of Fishing Capacity (IPOA-Capacity) encourages addressing this problem by utilizing capacity management that aligns fishing capacity with the sustainable use of their fish stocks (FAO) [3]. Therefore, the analysis and evaluation of fishing capacity is an important subject and task of supervision in offshore capture fisheries.

Fishing capacity is usually considered by fishery scientists in terms of effort and resulting fishing mortality. Fishing effort includes all the inputs used in the harvesting process. In practice, it is generally not possible to measure all inputs; thus, proxy measures are used such as the total days fished and the number of pots or kilometers of nets deployed. Fisheries managers generally have a similar view of fishing capacity, but they often link the concepts of capacity more directly to the number of fishing vessels in a fishery. This view is particularly common when the fishery is managed using input controls, because fleet size and effort level are the main control variables [4]. Therefore, fishing capacity usually considers measures related to gross tonnage. For example, it expresses total effort as a specific capacity metric multiplied by the number of fishing days per ship in the entire fleet. FAO (2000) suggested that fishing capacity may be defined with reference either to fishing inputs (vessels and potential effort) or to fishing output (potential catch) [5].

Quantitative methods for measuring fishing capacity include the following: rapid appraisal (RA) techniques, peak-to-peak analysis, stochastic production frontier (SPF), and data envelopment analysis (DEA). Moreover, biological, bioeconomic, and multi-objective modelling has been developed and widely used [6–8]. Peak-to-peak analysis is a relatively simple method that compares catch rates in different time periods and estimates potential catches based on peak catch rates on either sides of the year examined. The minimum information requirement is a time series of total output (i.e., total catch) and total inputs (e.g., days fished or vessels numbers). Although it is simple to calculate, it does not allow changes in stock conditions. Both DEA and SPF are frontier-based methods. That is, they are based on estimating the production possibility frontier, which is the maximum level of output that might be expected given a set of inputs. These can be used for the estimation of both capacity utilization and technical efficiency. The techniques require catch and input information on individual vessels, and they can be used to estimate the potential catch of each vessel separately. This requires more detailed information than that required by peak-to-peak analysis but provides a more reliable estimate. The DEA and SPF estimates of capacity are estimated using different procedures. DEA is a (non-parametric) linear programming-based approach, whereas SPF is a (parametric) statistical based approach. The methods recommended by countries such as Canada, Denmark, and the United States mainly include SPF and data envelopment analysis (DEA) [9].

There have been many scientific studies applying the above methods to assess fishing capacity. Vestergaard used the DEA-Malmquist model to analyze and study fishing capacity utilization of the Danish gillnet fleet [10]. In order to analyze the offshore fishing input by the data of the *China Fishery Statistical Yearbook* from 1979 to 2016 by using the DEA method, they believed that since the implementation of China's "dual control policy" for fishing vessels, it has not achieved the expected results due to many factors [11]. Tingley used the DEA-C2R model and SPF method to conduct a comparative study on the technical efficiency of the fleet. They found that DEA is more adaptable than SPF under certain circumstances [12]. Zheng used DEA to analyze the utilization of offshore fishing and pelagic fishing capacity in China from 1994 to 2005 [13]. Rao used DEA to analyze the fishing capacity of the East China Sea, the Yellow Sea, and the South China Sea [14]. Vassdal used the DEA-BC2 model to study the capacity utilization of the Norwegian Atlantic salmon fishing fleet from 2001 to 2008 [15]. Lim used a DEA-Malmquist model and SPF method to analyze the impact of the installation of echo sounders in Malaysian trawl and seine fleets on fishing technical efficiency [16]. Liang used a stochastic frontier analysis method to study the impact of multiple fishery control policies on the efficiency of the country's offshore fishing technology, which is based on the offshore fishing data of 11 provinces in China from 2008 to 2011 [17]. Asche used the DEA-Malmquist model to study the changes in total factor production efficiency of Norwegian salmon aquaculture enterprises [18]. Jo used the DEA method to study the impact of the technological development of trawlers from 1960 to 2010 on fishing capacity in Korea [19].

For fishery managers, the forecast of annual catches is very important. With the development of computer technology in recent years, machine learning algorithms have been

widely used. In particular, a regression analysis incorporating machine learning methods to predict the dependent variable by an optimal combination of multiple independent variables was well developed. Currently, a few studies on the application of machine learning regression method in marine fishery exist, but the method is widely used in agriculture, enterprise production, and other fields [20,21].

Using DEA can effectively estimate input and output efficiency values. However, it is affected by certain factors in DEA-calculated output values (annual catch), which is quite different from the actual annual catch value. Regression analysis using machine learning algorithms can more accurately reflect the trend of the output value (annual catch). The DEA method is used to analyze the utilization degree of the input factors of China's offshore fishing capacity, and the utilization rate of each input factor index can be obtained. On this basis, the machine learning algorithm is used to establish a regression model of the annual catch in China's offshore waters, which can predict the annual catch in the future. For this reason, the respective advantages of machine learning regression and DEA are combined. The fusion of the two methods can realize the calculation of the reasonable input value. It can also improve the accuracy of fishing yield forecasts, and the obtained values can also be mutually revised.

Management methods include resource management, total amount control and quota management, fishing moratorium, and reducing fishing vessel plans, restrictions, and fishing gear, etc. The management measures of each country are different, and this is due to the country's fishery resources distribution, number of fishermen and fishing vessels, and economic and social development levels of science and technology and decisions, such as national differences in various aspects. As a result, the conclusions of various studies cannot be completely copied but can only be referred to, and the results of each study can be supported according to the characteristics of their own country.

The purpose of this study is to analyze the utilization status of input indicator factors in China's offshore fishery in the past 17 years. At the same time, the study aims to explore the procedure for annual catch prediction by using a regression model combined with the machine learning algorithm. The relevant management policies of China's offshore fishery are analyzed and discussed. In this paper, DEA and fitting regressions were combined in order to comprehensively investigate the management policies adapted to the characteristics of Chinese fishery situation in order to provide support for the management of the fishery industry, the utilization statuses of each factor and annual catch change were considered simultaneously.

This mechanism can help fishery managers in formulating plans for national and regional fishing capacity management actions and provide optimization references for the next step in offshore fishing management decisions.

## 2. Materials and Methods

### 2.1. Literature and Research Structure

#### 2.1.1. Literature

Combined with the research literature of the preface, DEA, stochastic frontier method, and regression method are conducive in carrying out relevant research in the fishery field internationally, which mainly includes the following: guiding the sustainable development of fishery population in terms of fishery resources and population [22–26]; studying the efficiency of fishing capacity of fishing vessels and fleets [27–33]; studying production efficiency, technical efficiency, and economic efficiency of fisheries [34,35]; studying quota and fishing effort [36,37]; and studying the performance of fishery enterprises and the capacity levels of fishermen [38,39]. Table 1 summarizes the relevant literature.

**Table 1.** Literature Summary.

| Category | Literature | Main Methods and Brief Content |
|---|---|---|
| Fishery resources and population | Grosskopf S. (2001)<br>Wade L. Griffin (2011)<br>Ashley M. Apel (2013)<br>Thi Duy Thanh Pham (2014)<br>S. Gómez (2020) | DEA method was used to calculate the production capacity of single and multi-species fisheries, analyze the sustainable yield boundary, examine economic surplus boundary and other issues, and guide the sustainable management of fish population. |
| Fishing vessel and fleet and fishing capacity | Niels Vestergaard (2003)<br>Vestergaard N. (2003)<br>Tingley, D. (2005)<br>Luc Van Hoof (2005)<br>EfthymiaVtsttsika (2008)<br>M.M. Oliveira (2009)<br>Vassdal, T. (2011)<br>Collier T. C. (2014)<br>Castilla-Espino D. (2014)<br>Daniel Quijano (2018)<br>Zhang, Z.L. (2018)<br>Jo, H.S. (2020)<br>Tunca S. (2021) | By means of DEA, DEA-Malmquist, and stochastic frontier analysis, the efficiency, sustainable level and capacity utilization rate of multi-species, and different types of fleets were evaluated and tested, and regulatory policies were discussed. |
| Technical production and Economic efficiency | Diana Tingley (2005)<br>Lim, G. (2012)<br>Vazquez-Rowe I. (2013)<br>Pham T. (2014)<br>Chun-Jie Li (2020)<br>Rao, X. (2016)<br>Liang, S. (2016) | SPF and DEA methods were used to carry out regression, and the main factors affecting fishery technical efficiency, production efficiency, economic performance, capacity efficiency, and inefficiency caused by the "captain effect" were analyzed. |
| Quotas and effort | Duy N. N. (2016)<br>Asche, F. (2013)<br>Pascoe S. (2013) | DEA was used to analyze quota management and random production frontier (SPF), and data envelopment analysis (DEA) was used to analyze fishing unit effort (CPUE); there was no difference among efficiency scores. |
| Fishery enterprise and fishermen | Quynh C. N. T. (2018)<br>JI, X. (2020)<br>Chia-Nan Wang (2021) | DEA and endogenous conversion regression models were used to analyze the degree of overcapacity at the level of the individual fishermen. The performance of fishery companies was analyzed, and the influence of operator participation in monitoring the level of overcapacity was investigated. |

By using the above analysis, existing research and application either use the DEA method or fitting regression method. Factor utilization and technical efficiency and so on were studied, including the lack of convergence analysis of annual catch changes. The DEA method accounts for most of the research attributed to the DEA method recommended by FAO. Therefore, most of the support for fishery management is reflected in a certain aspect (such as fishing vessel, population, and technical efficiency and so on). The annual catch model is obtained by using a fitting regression algorithm. The factor utilization rate is studied by DEA, and the parameter values calculated by DEA are brought into the catch model for verification. Combined with the two results, the management of offshore fishing in China was analyzed. This study uses the integrated application of DEA and regression and conducts comprehensive investigation into factors while predicting the amount of annual fishing; the advantage of the combination is reflected. At the same time, the comprehensive analysis of the measures of China's offshore fishery management is discussed by using the obtained results; this is the special feature and innovation of this study. The results offer help for the next steps in fishery management decision making and optimization.

### 2.1.2. Research Structure

Initially, we explained the source and collection of data. In this study, DEA was used in fishing efficiency evaluation, and input and output analysis on fishery management was conducted. We compared four regressions and chose the better regression model. Furthermore, DEA and improved regression models were used for data simulation. Finally, fishery management plans and strategies were proposed based on this. The flowchart of the methodology used in this paper is shown in Figure 1.

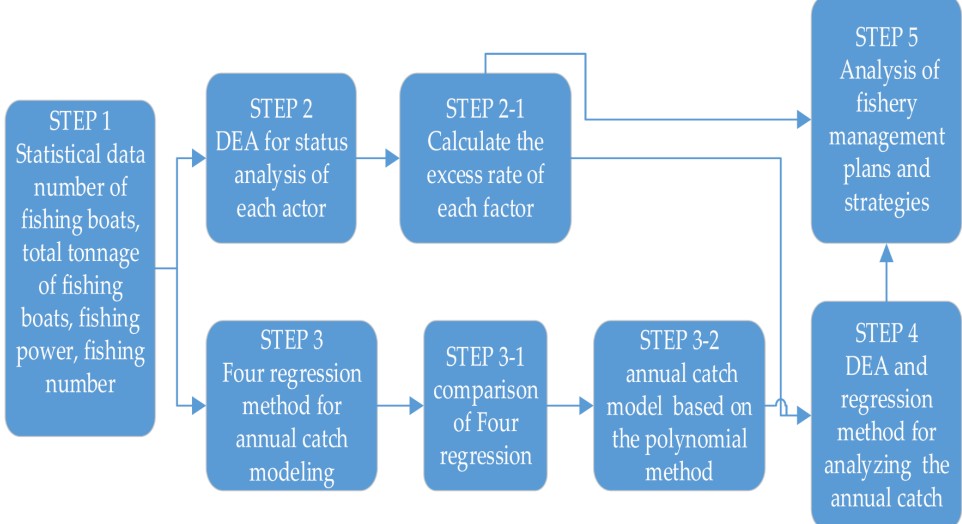

**Figure 1.** The method flow chart of this paper. Section 2.2 Material and Data.

The data come from the data on offshore fishing operations in China obtained from the *China Fishery Statistical Yearbook* published from 2004 to 2020. Related factor data [40], including the North Sea, Yellow Sea, East China Sea, Taiwan Strait, the South China Sea, inland sea, coastal sea, adjacent sea, and the exclusive economic zone and other fishery operations areas, are collectively referred to as offshore operations areas and offshore marine fishing data. At the same time, the data of the "National Fishery Statistics Data Website" was compared [41]. The annual offshore fishing data from 2003 to 2019 were extracted, including the total number of fishing vessels (ship), total ton of fishing vessels (ton), total power of fishing vessels (kw), professional fisherman (number), and annual catch (t) (Table 1), which are detailed in the records and statistics of the *China Fishery Statistical Yearbook*.

By taking the number of fishing vessels, total tonnage of fishing vessels, total fishing boat power, and professional fishing labor as input measures and annual catch as output measure, the data in Table 2 were compiled.

From Figure 2, it can be observed that (1) the number of fishing vessels and fishermen has experiencing a downward trend, and the power of fishing vessels has shown a upward trend and downward trend after 2015, which reflects the effect of China's policy. (2) Although the power of fishing vessels has been reduced, the total tonnage of fishing boats has shown an upward trend. According to the fishing boat design theory, in order to obtain a certain sailing speed, fishing boats will have large machines and small standards in the fishing industry. (3) With the rising trend of total tonnage, there is also a reduction in fishing boats. Small and medium-sized fishing boats account for a relatively large proportion, while large-scale fishing boats have a relatively small reduction. (4) Since 2013, the number of offshore fishers has shown a downward trend, reflecting the effect of the implementation of the policy of conversion of fishermen. (5) The annual catch of offshore fishery has shown a downward trend year by year. In 2019, the catch was 10 million tons, achieving the goal of controlling the annual catch of offshore fisheries within 10 million tons by the end of 2020.

**Table 2.** China's offshore fishing data from 2003 to 2019.

| Year | Total Number of Fishing Vessels (Ship) | Total ton of Fishing Vessels (ton) | Total Power of Fishing Vessels (kw) | Professional Fishermen (Number) | Annual Catch (t) |
|---|---|---|---|---|---|
| 2003 | 224,843 | 5,635,555 | 12,384,435 | 1,148,298 | 14,323,121 |
| 2004 | 220,342 | 5,559,435 | 12,338,132 | 1,119,726 | 14,510,858 |
| 2005 | 214,560 | 5,547,974 | 12,363,850 | 1,102,630 | 14,532,984 |
| 2006 | 211,314 | 5,463,309 | 12,433,501 | 1,076,206 | 14,160,007 |
| 2007 | 207,353 | 5,527,675 | 12,394,224 | 1,074,398 | 11,360,329 |
| 2008 | 199,949 | 5,776,472 | 12,950,657 | 1,073,879 | 11,496,270 |
| 2009 | 206,923 | 5,838,599 | 13,058,326 | 1,071,080 | 11,786,109 |
| 2010 | 204,456 | 6,010,919 | 13,040,623 | 1,066,329 | 12,035,946 |
| 2011 | 201,694 | 6,182,268 | 13,255,855 | 1,032,175 | 12,419,386 |
| 2012 | 194,240 | 6,517,469 | 13,270,770 | 1,054,191 | 12,671,891 |
| 2013 | 196,803 | 6,887,624 | 13,614,004 | 1,089,526 | 12,643,822 |
| 2014 | 191,944 | 7,294,059 | 14,087,583 | 1,060,566 | 12,808,371 |
| 2015 | 187,211 | 7,572,484 | 14,417,390 | 1,025,807 | 13,147,811 |
| 2016 | 179,688 | 7,684,810 | 14,308,795 | 1,002,122 | 11,872,029 |
| 2017 | 166,349 | 7,649,188 | 13,782,628 | 990,325 | 11,124,203 |
| 2018 | 156,018 | 7,820,086 | 13,701,490 | 960,345 | 10,444,647 |
| 2019 | 146,951 | 7,917,408 | 13,547,244 | 921,283 | 10,001,515 |

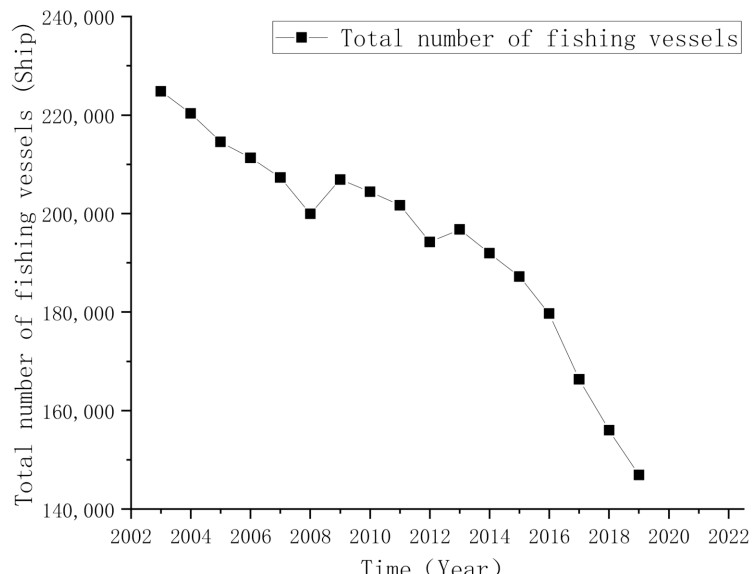

(**a**) Total number of fishing vessels, 2003–2019.

**Figure 2.** *Cont.*

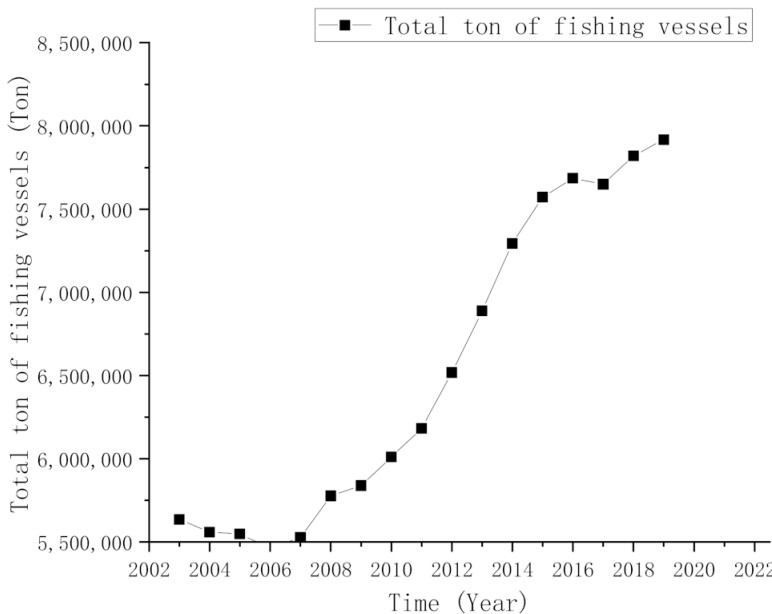

(**b**) Total tonnage of fishing vessels, 2003–2019.

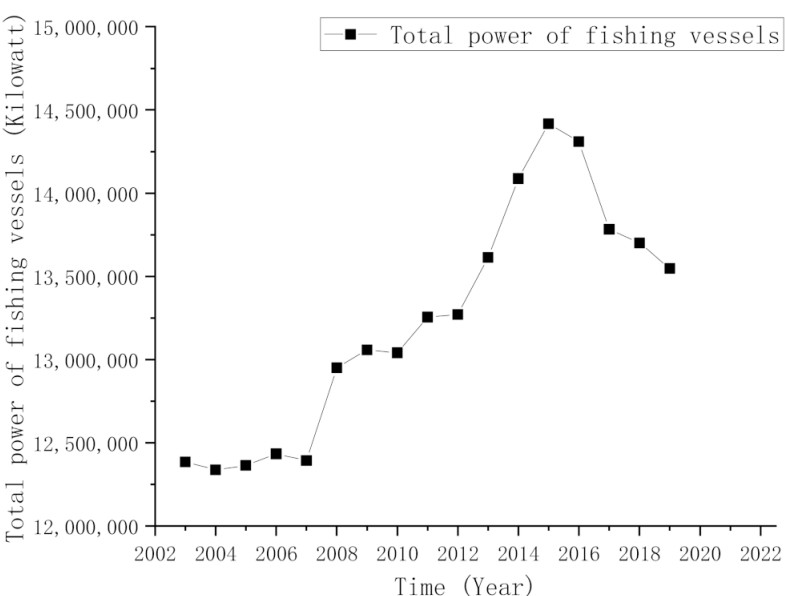

(**c**) Total power of fishing vessel, 2003–2019.

**Figure 2.** *Cont.*

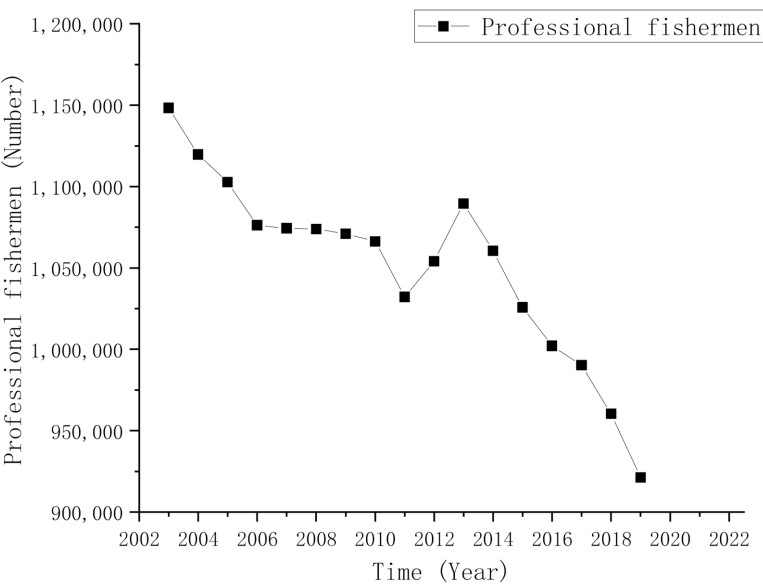

(**d**) Total number of fishermen, 2003–2019.

**Figure 2.** Four factors fluctuating in China's offshore fisheries from 2003 to 2019.

*2.2. Methods*

2.2.1. Data Development Analysis (DEA)

Data envelopment analysis (DEA) is a mathematical programming approach for estimating the relative technical efficiency (TE) of production activities. The term DEA was originally proposed by Charnes et al. [42]. Since the early study of Charnes et al., DEA has been developed and expanded to include a wide variety of applications. DEA can conduct input and output analysis on different issues in a targeted manner. Multi-layer system models have been developed, including C2R, C2W, C2WH, and C2GS2 models [43,44]. Among them, the C2R model was the first model widely used in marine fisheries assessment and was recommended by the World Food and Agriculture Organization [45].

It is assumed that there are M decision-making units (DMU) from the C2R model. Each unit contains G kinds of inputs and N kinds of outputs. Then, the quantity of the first DMU is expressed as $x_i$ and $y_i$, and the input data and output data of the DMU can be formed into a G*K matrix and a N*K matrix separately. These two matrices are denoted as *X* and *Y* [46]. In addition, if the return to scale is set as a constant (CRS), a linear programming model for the *i*-th DMU can be established.

$$
\begin{aligned}
&Min_{\theta,\,\lambda}\theta \\
&s.t. \\
&-y_i + Y\lambda \geq 0 \\
&\theta x_i - X\lambda \geq 0 \\
&\lambda \geq 0
\end{aligned}
\tag{1}
$$

Among them, $\lambda$ is a constant vector of $N \times 1$, and $\theta$ is the utilization rate of the *i*-th DMU input ($0 \leq \theta \leq 1$). If $\theta = 1$, the utilization rate is 100%. On this basis, the surplus rate of investment in the literature is used for calculations [13]. The DEA algorithm uses the optimal output as a reference and calculates the minimum theoretical input value required when the catch in each year remains unchanged. Based on this output result, the excess rate of a certain input variable in that year can be obtained. The specific calculation method is described as follows.

$$
Excess\ (\%) = \frac{Actual\ input\ value - Input\ direction\ value}{Actual\ input\ value} \times 100\%
\tag{2}
$$

The DEAP2.0 software is used in the calculation process, and the multi-part method is used to deal with slack variables.

### 2.2.2. Regression Analysis Incorporating Machine Learning

Regression analysis that incorporate machine learning algorithms has been widely used, such as Lasso, Ridge, KNN, and Polynomial Features. Among them, Lasso regression is a biased estimation method for dealing with multicollinearity data. Ridge regression is one of the most commonly used methods for studying collinearity data, which is modified from the least square method and is a method of partial estimation regression. The KNN algorithm is a classic data processing method of machine learning, which has the advantages of no estimation and no training. The Polynomial Features method can have a good effect in the field of predicting the outcome of a dependent variable based on the characteristics of the independent variable of the data [47–49].

#### Lasso Regression

The Lasso method is a regression analysis method that performs both variable selection and regularization in order to enhance prediction accuracy and interpretability of the resulting statistical model [50]. It is characterized by least absolute shrinkage and selection operator, which has been widely used in statistics and machine learning. Although originally defined for linear regression, lasso regularization is easily extended to other statistical models including generalized linear models, generalized estimating equations, proportional hazards models, and M-estimators [51,52].

The idea that Lasso uses for estimation is based on the cross-validation method proposed by Efron and Tibshirani in 1993 [53]. The specific description is described as follows.

Set $p$ independent variables, $x_1, x_2, \ldots, xp$, and dependent variable $y$; the model is defined as follows:

$$y = \alpha + \beta_1 x_1 + \beta_2 x_2 + \ldots + \beta_p x_p + \varepsilon \tag{3}$$

where $a$ is a constant term, $\beta_1, \beta_2, \ldots \beta_p$ is a regression coefficient, and $\varepsilon$ is a random error term.

Suppose $(x_{i1}, \ldots x_{in}, y_i)$; $i = 1, 2, \ldots, n$ are the n sets of observations of variables. Existing data were centralized and standardized as follows.

$\sum_{i=1}^{n} y_i = 0$, $\sum_{i=1,j=1}^{n} x_{ij} = 0$, $\sum_{i=1,j=1}^{n} x_{ij}^2 = 1$, $j = 1, 2, \ldots, p$, then $\beta = (\beta_1, \beta_2, \ldots, \beta_p)^Y$.

The parameters $\alpha$ and $\beta$ are defined by Lasso's estimation.

$$(\hat{\alpha}, \hat{\beta}) = \text{argmin} = \left\{ \sum_{i=1}^{n} \left( y_i - \alpha - \sum_{j=1}^{p} \beta_j x_{ij} \right)^2 \right\} \tag{4}$$

They also obey $\sum_{j=1}^{p} |\beta_i| \leq s$.

Among them, $s \geq 0$ is the penalty parameter. $\hat{\beta}^0$ is the least square solution, $s_0 = \sum_{i=1}^{p} |\beta_i|$.

When $s > s_0$, the equation becomes the following: $(\hat{\alpha}, \hat{\beta}) = \text{argmin} = \left\{ \sum_{i=1}^{n} \left( y_i - \alpha - \sum_{j=1}^{p} \beta_j x_{ij} \right)^2 \right\}$.

The optimal solution obtained in the equation is a least square solution.

When $s < s_0$, some regression coefficients shrink or approach zero or will even equal to zero. Variables equal to zero are eliminated, and then the variable is chosen.

For standardized and centralized data, for any $s > 0$, Equation (4) is stated as follows.

$$(\hat{\alpha}, \hat{\beta}) = \text{argmin} = \left\{ \sum_{i=1}^{n} \left( y_i - \alpha - \sum_{j=1}^{p} \beta_j x_{ij} \right)^2 \right\}.$$

$\hat{\alpha} = 0$; the original formula can be changed to the following.

$$\hat{\beta} = \text{argmin} = \left\{ \sum_{i=1}^{n} \left( y_i - \alpha - \sum_{j=1}^{p} \beta_j x_{ij} \right)^2 \right\} \tag{5}$$

For all values $s \geq 0$, a Lasso solution is obtained by using the above equation, and the difference can be obtained after calculation. All Lasso solutions are under the s value.

The advantage of Lasso is to add the penalty limit of $\sum_{j=1}^{p} |\beta_i| \leq s$. By compressing the coefficients of some meaningless or extremely small independent variables to zero, more meaningful independent variables are screened out, and the model determination coefficient R2 is larger [54].

Lasso can also be expressed by the minimum of the residual sum of squares plus a penalty function for regression coefficients [55].

$$\min_{\beta} \sum_{i=1}^{n} \left( y_i - \sum_{j=1}^{p} \beta_j x_{ij} \right)^2, s.t \sum_{j=1}^{p} |\beta_j| \leq \lambda \tag{6}$$

The estimated coefficient of this method is described as follows.

$$\hat{\beta}_{\text{lasso}} = \arg \min = \left\{ ||y_i - \sum_{j=1}^{p} X_{ij} \beta_j||^2 + \lambda \sum_{j=1}^{p} |\beta_j| \right\}, s.t. \sum_{j=1}^{p} |\beta_j| \leq \lambda \tag{7}$$

Among them, $\lambda$ is the adjustment parameter ($\lambda > 0$). As $\lambda$ increases, the $\sum_{j=1}^{p} |\beta_i|$ term decreases. At this time, the coefficients of some independent variables must be gradually compressed to zero in order to reduce dimensionality of high-dimensional data.

Ridge Regression

Ridge regression is a biased estimation regression method dedicated to collinearity data analysis (Hoerl and Kennard, 1970) [53,54]. Its characteristic is to modify the estimated coefficient derived from this regression based on the least squares principle in order introduce more stability. In order to reduce variance and to introduce some bias, this method is called regularization, which is beneficial for the predictive performance of the model. Although some information is lost and accuracy is reduced, the regression coefficients obtained are more realistic and more reliable. Normally, the fit to ill-conditioned data is stronger than the least square method. Ridge regression is a frequently used method when the independent variables are highly correlated. Therefore, it is widely used in many fields, including econometrics, chemistry, engineering, and fisheries.

When multicollinearity occurs between the independent variables of the equation, it is "$|X'X| \approx 0$." In the process of obtaining parameters in multiple regression models, the stability of parameter $\hat{\beta} = (|X'X|_{-1})X'Y$ is very poor. At the same time, the mean square error of the β estimator, $\hat{\beta}$, becomes very large when the least square estimation method is used. Therefore, the ridge regression method standardizes data in order to obtain design matrix $X$, and its expression is as follows:

$$\beta(k) = (X'X + kI_m)^{-1} X'Y$$

where $k$ is the ridge parameter; that is, a non-negative factor $k$ was added to the main diagonal element of the matrix $X'X$, where $Im$ is the unit matrix of order $m$, and $k > 0$ is

called the collar parameter or the partial parameter. If $k$ takes a constant that is unrelated to experimental data $Y$, then $\beta(k)$ is a non-linear estimation.

KNN Regression

The K-Nearest Neighbors algorithm (KNN) is a lazy learning, non-parametric algorithm and is one of the most used learning algorithms. With respect to KNN regression, its essence is that it approximates the correlation between independent variables and continuous results by averaging observations in the same neighborhood. The distance function between the sample to be classified, $x$, and each training sample was calculated by KNN regression. The K samples are selected for the smallest distance from the sample to be classified as the K nearest neighbors of $x$ and finally judges the category of $x$ based on the K nearest neighbors of $x$ (Cover and Hart, 1968) [55]. The distance formulaa in Euclidean two-dimensional and three-dimensional space are described as follows.

Two-dimensional: $\rho = \sqrt{(x_2 - x_1)^2 + (y_2 - y_1)^2}$, $\left|X\right| = \sqrt{x_2^2 + y_2^2}$;

Three-dimensional: $\rho = \sqrt{(x_2 - x_1)^2 + (y_2 - y_1)^2 + (z_2 - z_1)^2}$, $\left|X\right| = \sqrt{x_2^2 + y_2^2 + z_2^2}$.

Among them, $\rho$ is the distance between points $(x_2, y_2)$ and $(x_1, y_1)$, and $X$ is the distance from $(x_2, y_2)$ to the origin.

Polynomial Features Algorithm

The Polynomial Features algorithm can also obtain good results for small samples data in order to study the influence of compound independent variables on dependent variables. It is a construction method used to express independent variables. The essence of this method is to use a polynomial method for data processing. Among them, polynomial regression can be transformed into multiple linear regressions by using variables. Let $x_1 = x$, $x_2 = x_2$, ... , $x_n = x_n$, then $y = b_0 + b_1 x + b_2 x_2 + \ldots + b_n x_n$ is transformed into an n-array linear regression equation: $y' = b_0 + b_1 x + b_2 x_2 + \ldots + b_n x_n$.

In the process of polynomial regression analysis, testing whether the regression coefficient bi is significant is conducted in order to judge whether the $i$-th power term of the independent variable x has a significant impact on the dependent variable $y$. By analogy, the multivariate quadratic polynomial regression equation is defined as follows. Let $z_{1,i} = x_{1,i}$, $z_{2,i} = x_{2,i}$, $z_{3,i} = x_{3,i}$, ... , $z_{k,i} = x_{k,i}$; in order to increase the regression accuracy of the regression equation, add $z_{1,i} = x_{1,i}$, $z_{2,i} = x_{2,i}$, ... , $z_{k,i} = x_{k,i}$, where $\varepsilon i$ is the residual term [56–60].

The regression equation is described as follows.

$y_i = b_0 + b_1 z_{1,i} + b_2 z_{2,i} + \cdots + b_k z_{k,i} + x_{1,i} + x_{2,i} + \cdots + x_{k,i} + \varepsilon_i$, where $(i = 1, 2, \ldots, n)$

Method Calculation Tool

The software environment comprises the Windows64 operating system, Pycharm20.1, Python3.8.8, torch 1.9.0+cu111, CUDA11.1, and CUDNN v8.0.5.

Model Identification Parameters

In this study, the above-mentioned four regression models were used to calculate the predicted value of the annual catch. At the same time, mean square error (MSE), mean absolute error (MAE), and the coefficient of determination R2 were used as indicators to measure the pros and cons of the model.

## 3. Results

### 3.1. Data Envelopment Analysis

Using DEA to calculate offshore fishing capacity in 17 years, the results are shown in Table 2. According to formula (1), the average capacity utilization rate is 93.5%, and the range is 80.7–100%. From 2007 to 2010, it showed low efficiency. The results are shown in Table 3.

**Table 3.** Calculation of fishing capacity in China from 2003 to 2019.

| Year | Actual Input Value | | | | | DEA Calculate the Value (Output Direction) | | | DEA Calculate the Value (Input Direction) | | |
|---|---|---|---|---|---|---|---|---|---|---|---|
| | Annual Production(t) A | Number of Vessels B | Total ton C | Total Power D | Number of Fisherman E | Capacity Utilization (%) F | Fishing Capacity Production (t) G | Number of Vessels H | Total ton I | Total Power J | The Number of Fisherman K |
| 2003 | 14,323,121 | 224,843 | 5,635,555 | 12,384,435 | 1,148,298 | 98.3 | 14,570,825 | 217,491 | 5,487,509 | 12,178,505 | 1,105,239 |
| 2004 | 14,510,858 | 220,342 | 5,559,435 | 12,338,132 | 1,119,726 | 100 | 14,510,858 | 220,342 | 5,559,435 | 12,338,132 | 1,119,726 |
| 2005 | 14,532,984 | 214,560 | 5,547,974 | 12,363,850 | 1,102,630 | 100 | 14,532,984 | 214,560 | 5,547,974 | 12,363,850 | 1,102,630 |
| 2006 | 14,160,007 | 211,314 | 5,463,309 | 12,433,501 | 1,076,206 | 99.8 | 14,188,384 | 209,053 | 5,405,590 | 12,046,542 | 1,074,332 |
| 2007 | 11,360,329 | 207,353 | 5,527,675 | 12,394,224 | 1,074,398 | 80.7 | 14,077,235 | 167,381 | 4,462,103 | 9,823,327 | 863,305 |
| 2008 | 11,496,270 | 199,949 | 5,776,472 | 12,950,657 | 1,073,879 | 84.2 | 13,653,527 | 168,437 | 4,866,102 | 10,384,671 | 877,518 |
| 2009 | 11,786,109 | 206,923 | 5,838,599 | 13,058,326 | 1,071,080 | 83.8 | 14,064,569 | 173,308 | 4,757,648 | 10,353,912 | 897,082 |
| 2010 | 12,035,946 | 204,456 | 6,010,919 | 13,040,623 | 1,066,329 | 86.2 | 13,962,814 | 176,237 | 5,134,336 | 10,922,545 | 919,152 |
| 2011 | 12,419,386 | 201,694 | 6,182,268 | 13,255,855 | 1,032,175 | 91.3 | 13,602,832 | 183,356 | 4,741,107 | 10,565,719 | 942,270 |
| 2012 | 12,671,891 | 194,240 | 6,517,469 | 13,270,770 | 1,054,191 | 94.5 | 13,409,408 | 183,517 | 6,157,657 | 12,451,590 | 976,045 |
| 2013 | 12,643,822 | 196,803 | 6,887,624 | 13,614,004 | 1,089,526 | 92.8 | 13,624,808 | 182,658 | 6,311,135 | 12,635,546 | 975,733 |
| 2014 | 12,808,371 | 191,944 | 7,294,059 | 14,087,583 | 1,060,566 | 95.7 | 13,383,878 | 183,595 | 6,926,402 | 13,474,826 | 994,334 |
| 2015 | 13,147,811 | 187,211 | 7,572,484 | 14,417,390 | 1,025,807 | 100 | 13,147,811 | 187,211 | 7,572,484 | 14,417,390 | 1025,807 |
| 2016 | 11,872,029 | 179,688 | 7,684,810 | 14,308,795 | 1,002,122 | 94.1 | 12,616,396 | 169,045 | 6,837,697 | 13,018,416 | 926,269 |
| 2017 | 11,124,203 | 166,349 | 7,649,188 | 13782,628 | 990,325 | 95.2 | 11,685,087 | 158,397 | 6,406,987 | 12,198,378 | 867,923 |
| 2018 | 10,444,647 | 156,018 | 7,820,086 | 13,701,490 | 960,345 | 95.3 | 10,959,756 | 148,721 | 6,015,596 | 11,453,203 | 814,903 |
| 2019 | 10,001,515 | 146,951 | 7,917,408 | 13,547,244 | 921,283 | 96.9 | 10,321,481 | 142,411 | 5,760,374 | 10,967,281 | 780,329 |
| | | | Average value: | | | 93.5 | | | | | |

The results of excess investment rates of various factors are obtained, including the total number of fishing vessels, gross tonnage, total power, and the total number of professional fishing (Figure 3). Regarding the trends of the four factors, there was a common trend during the period from 2003 to 2015. However, after 2015, the total number of fishing vessels showed a decline differently from the other three. During the first four years of this period, excess investment rate was low (<5%). There was a sharp upward trend in 2007, then a gradual downward trend from 2007 to 2015, reaching a low point in 2015, and then an upward trend, with the highest gross tonnage of fishing vessels reaching 25.5%.

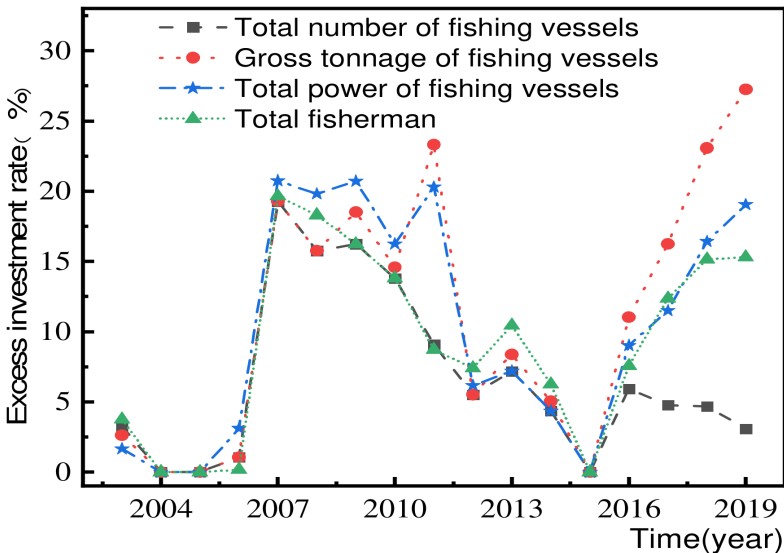

**Figure 3.** The fluctuation of excess investment rate from 2003 to 2019.

### 3.2. Comparison of Four Regression Models

Four prediction curves of the annual catch were estimated using Polynomial Features, Lasso, Ridge, and KNN methods, respectively. Ridge and KNN are derived from the least square method, and they show the same trend. Both real data and predicted data are plotted and used for comparison, as shown in Figures 4–7.

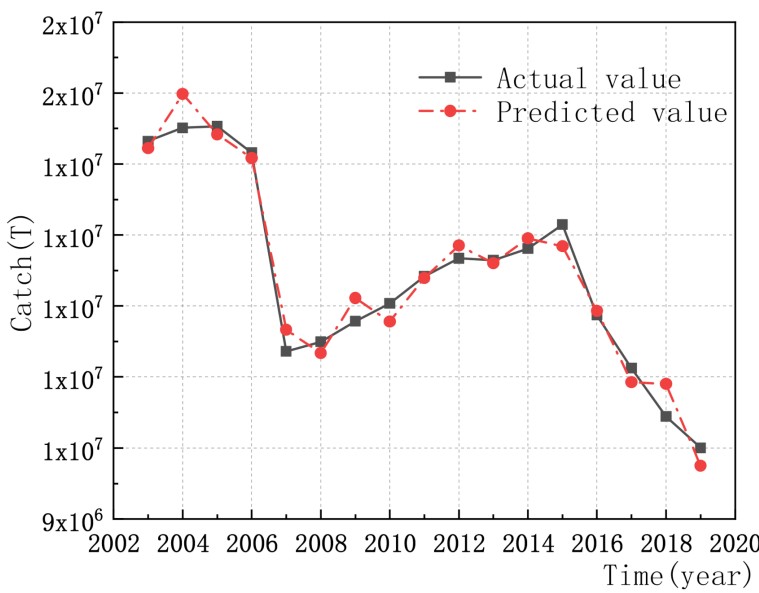

**Figure 4.** Polynomial regression.

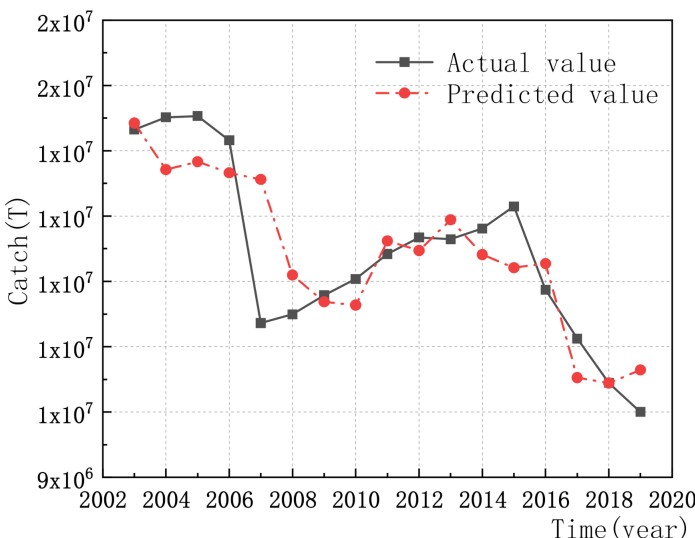

**Figure 5.** Lasso regression.

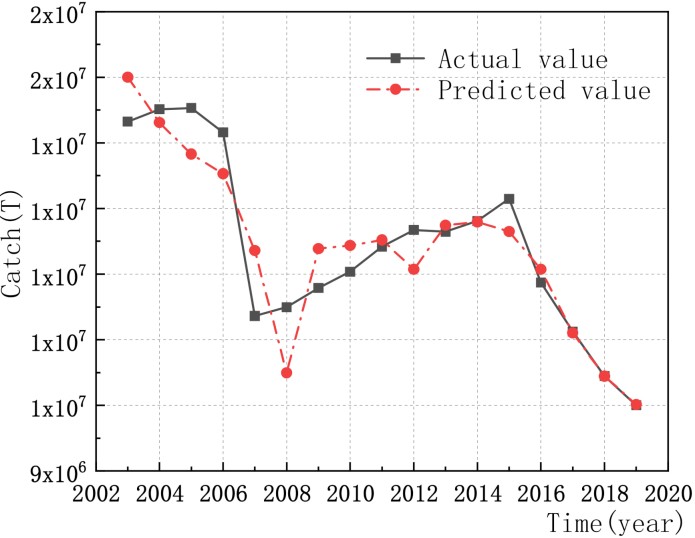

**Figure 6.** Ridge regression.

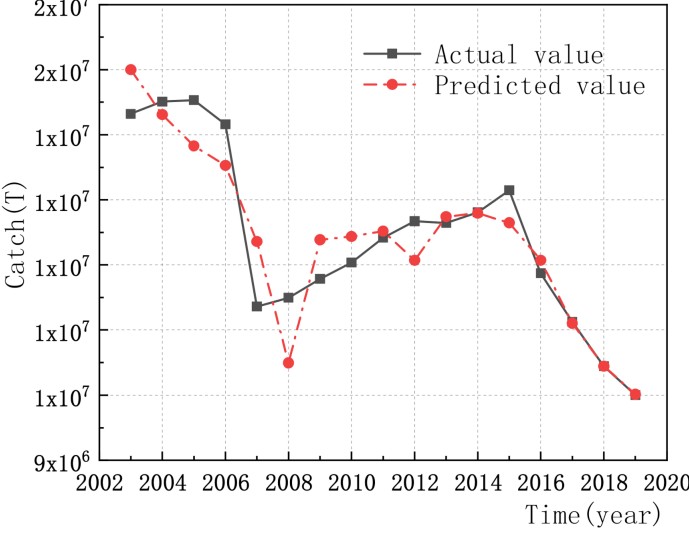

**Figure 7.** KNN regression.

According to Figures 4–7, all four predicted curves can effectively reflect the trend of true data. In order to compare the goodness of fit of the four models, average absolute error, mean square error, and the coefficient of determination ($R^2$) were calculated respectively. The results are shown in Table 4. The Polynomial model performed significantly better than the other three models. Therefore, it shows that it has more advantages in forecasting annual catches.

**Table 4.** Comparison of four regression models.

| Model | Average Absolute Error (MAE) | Mean Square Error (MSE) | Coefficient of Determination ($R^2$) |
|---|---|---|---|
| Polynomial model | 981,481.421 | 600,119.141 | 0.984 |
| Lasso model | $3.156 \times 10^6$ | $2.389 \times 10^6$ | 0.813 |
| Ridge model | $3.156 \times 10^6$ | $2.389 \times 10^6$ | 0.813 |
| KNN model | $3.918 \times 10^6$ | $2.921 \times 10^6$ | 0.721 |

In short, the Polynomial model is better than the other models. A regression model for the annual catch of China's offshore fishery was established based on the Polynomial model, as shown in formula (8).

$$y = -0.077x_0^2 - 0.006x_0x_1 - 0.001x_0x_2 + 0.002x_0x_3 + 76371.773x_0 - 1.67 \times 10^{-4}x_1^2 + 1.41 \times 10^{-4}x_1x_2 + 3.38 \times 10^{-4}x_1x_3 + 1061.77x_1 - 1.376x_2^2 + 5.98 \times 10^{-4}x_2x_3 - 933.856x_2 + 0.005x_3^2 - 21008.513x_3 + 6.288e + 09 \qquad (8)$$

Among them, $y$ represents the annual catch; $x_0$ represents the number of fishing vessels; $x_1$ represents the total tonnage of fishing vessels; $x_2$ represents the total power of fishing vessels; and $x_3$ represents the total number of professional fishermen.

### 3.3. Combination of Polynomial Model and DEA Method

If the amount of input factors is reduced, the total number of fishing vessels would be reduced by 10,000, the total tonnage of fishing vessels would be reduced by 200,000 tons, the total power of fishing vessels would be reduced by 400,000 KW, and the number of professional fishermen would be reduced by 40,000. The annual catch by using the Polynomial model was calculated, and the results are shown in Table 5. By using the above-mentioned DEA method, the degrees of input, capacity utilization, and the amount of catch output are calculated. The above values were substituted into the DEA method for efficiency calculation, and the results are shown in Table 6.

**Table 5.** Calculation of Polynomial model.

| Time | Total Number of Fishing Vessels (Ship) | Total ton of Fishing Vessels (t) | Total Power of Fishing Vessels (kw) | Professional Fishermen (Number) | Annual Catch (t) |
|---|---|---|---|---|---|
| 2019 | 136,951 | 7,717,408 | 13,147,244 | 881,283 | 9,554,489 |

Furthermore, the values of the total number of fishing vessels, gross tonnage, total power, and number of professional fishermen are calculated by the DEA method. Then, these data are used as independent variables of the Polynomial model to calculate the forecasted annual catch in offshore waters. This is the annual catch under the condition of 100% utilization, as shown in Table 7.

**Table 6.** Calculation of DEA.

| Year | Assume Actual Input Value | | | | | | The Value of DEA (Output Direction) | | The Value of DEA (Input Direction) | | |
|---|---|---|---|---|---|---|---|---|---|---|---|
| | Annual Production (t) A′ | Number of Vessels B′ | Total ton C′ | Total Power D′ | Number of Fishermen E′ | Capacity Utilization (%) F′ | Fishing Capacity Production(t) G′ | Number of Vessels H′ | Total ton I′ | Total Power J′ | The Number of Fishermen K′ |
| Assume a value | 9,554,489 | 136,951 | 7,717,408 | 13,147,244 | 881,283 | 95.4 | 10,015,187 | 121,377 | 5,788,056 | 10,649,267 | 731,464 |
| | | | Average value: | | | | 95.4 | | | | |

**Table 7.** Prediction of characteristic Polynomial model.

| Time | Total Number of Fishing Vessels (Ship) | Total ton of Fishing Vessels | Total Power of Fishing Vessels (kw) | Professional Fishermen (Number) | Annual Catch of Model (t) |
|---|---|---|---|---|---|
| 2019 | 121,377 | 5,788,056 | 10,649,267 | 731,464 | 9,053,687 |

In summary, the combined application of the Polynomial model and the DEA method enhanced the analysis effect of offshore fishing capacity. The combined applications can improve the accuracy of fishing output prediction while obtaining a reasonable investment value for fishing capacity. It also provides a reference for the pre-judgment of the next regulatory measures.

## 4. Discussion

### 4.1. Data Envelopment Analysis

The DEA method proposes two solutions to the problem of slack variables of input and output: "two-step method" and "multi-step method". Although the multi-step method is more complicated in processing difficulty, it minimizes the sum of slack variables when dealing with slack variables. Moreover, it does not cause changes in the optimal solution value due to changes in the measurement units of input and output. Based on the above reasons, the "total number", "total power", "gross tonnage", and "professional fishermen" of offshore fishing vessels as input independent variables are used. With "annual catch" as the output dependent variable, the slack variable uses the degree step method for data evaluation and analysis. The parameters and model selection of DEA used in this paper refer to the methods of [13,14,44]; thus, the results obtained are correct and feasible [14].

The difference is that the statistical data used in the study [13] is dated up to 2005, that is, data up to China's "Tenth Five-Year Plan" period. The statistical data used in the literature [14] id dated during the period of the "Twelfth Five-Year Plan" in 2014. In existing research, the statistical data used lacked summary and analysis of China's offshore fishing capacity during the 13th Five-Year Plan period. The statistics used in this article are dated up to 2019. In other words, at of the end of "13th Five-Year Plan", 2021 is the first year of the "14th Five-Year Plan". We use complete statistical data to summarize and analyze China's offshore fishing capacity during the 13th Five-Year Plan and before, which will help accumulate excellent experience and results.

From 2003 to 2019, the average utilization of the four input factors was 93.5%. It shows that the four indicators are all in excess. Since 2016, the surplus rate of the number of fishing vessels has been brought under control and has shown a downward trend. The surplus of gross tonnage and total power of fishing vessels is relatively large, and the surplus of gross tonnage shows an upward trend; the upward trend is greater than the total power of fishing vessels. The number of professional fishermen is also on the rise, but it has been relatively stable since 2018. For this reason, we suggest that corresponding implementations should be taken in the next step in order to control and reduce the four investments.

### 4.2. Comparison of Four Regression Models

The comparison of four regression models is shown in Figures 4–7. We think that the trends obtained among the four models are consistent with the true catch trends, indicating that the four models are appropriate. According to Table 4, the range of the coefficient of determination R2 is between [0,1]. The value of R2 is close to 1, which means that the regression effect of the model is better and vice versa. Comparing the effects of the four models, R2 of the Polynomial model is 0.984, and it is much better than the Lasso model (0.813), Ridge model (0.813), and KNN model (0.721). Moreover, it is higher than the model predictive index 0.86. By comparing the mean absolute error (MAE) and mean square error (mse) of the four models, the mean square error of the Polynomial model is much smaller than the other three methods. Therefore, the regression model established by the

Polynomial method has a better fitting ability. However, the Polynomial model seems to be closer to the real value, showing its superiority.

The results of DEA theoretical calculations obtained from formula (8) are shown in Figure 8a for comparison purposes. The curves of the Polynomial model and real values are shown to be closer than the curve of DEA theoretical calculations. The other comparison is shown in Figure 8b, the difference between the actual results and Polynomial model is counted as Z1, and the difference between the actual results and DEA theoretical calculation is counted as Z2. The average of Z1 and Z2 is also counted. It shows that the difference in the DEA theoretical calculation seems larger than the difference in the Polynomial model. This also verifies the effectiveness and superiority of the model. We believe that the use of the Polynomial model can provide help and reference for subsequent calculations and predictions of annual catches in offshore fisheries.

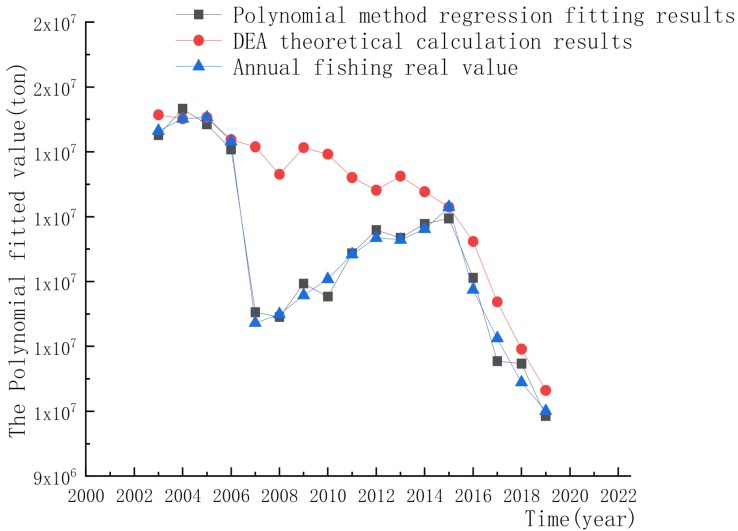

(**a**) Contrast figure of the Polynomial model and DEA theoretical calculations

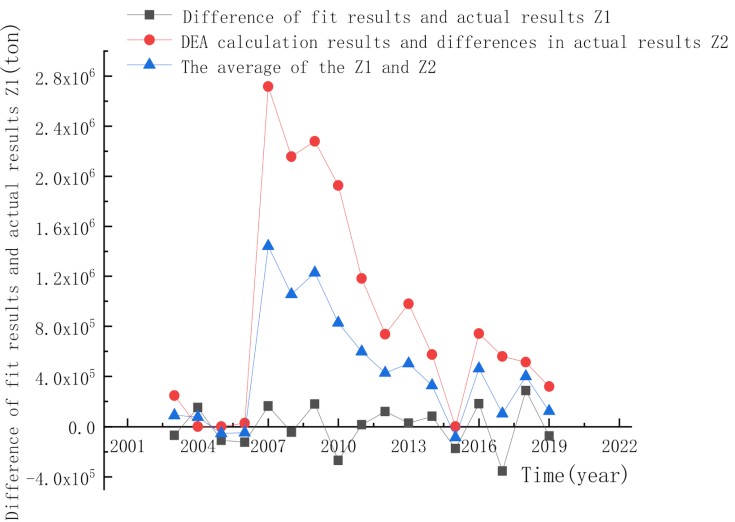

(**b**) Contrast figure of DEA, Polynomial with actual results

**Figure 8.** The contrast figure of data envelope and the fitting model with actual results.

### 4.3. Combination of Polynomial Model and DEA Method

The DEA method is often used to compare the input measure and output measure between different decision-making units. However, DEA also has the limitations of anti-interference ability and low accuracy of estimating dependent variables [13].

Regression analysis can perform accurate prediction. As the data sample size increases, the regression method can continuously optimize prediction accuracy by using algorithms. Under the given range conditions, regression analysis only reflects the relationship of the independent variable on the dependent variable and cannot evaluate and analyze input factors [53–56]. In this study, the results show that the DEA method can evaluate the utilization status of input factors and calculate the excess rate of each input factor under the conditions of a given range. We believe that it has more advantages for analyzing offshore fishing capacities while the DEA in combination with machine learning regression analysis.

In this paper, a simulation was conducted based on the data given; the total number of fishing vessels is 136,951, the total tonnage of fishing vessels is 7,717,408, the total power of fishing vessels is 13,147,244 kilowatts, and the number of professional fishermen is 881,283 (Table 5). By using the Polynomial model, the annual catch was calculated to be 9,554,489 tons. Then, the above values are used as inputs and are substituted into the DEA method for calculation. We set the average utilization rate of input factors to 95.4%, and the results are shown in Table 6. If the utilization rate of the above factors reaches 100%, the theoretical values are calculated repeatedly by the DEA, and the number of fishing vessels available is 121,377, the gross tonnage is 5,788,056, the total power of fishing vessels is 10,649,267 kilowatts, and the number of professional fishermen is 731,464. Then, this parameter is brought into the Polynomial model, and the available annual catch is 9,053,687 tons. Based on the above calculations, the simulated annual catch is 9.0537 million tons, as shown in Table 7.

By comparing the input factors two times, they are able to reduce the total number of fishing vessels by 11.4%, the total tonnage to 25.0%, and the power to 19.0%. Using DEA incorporating regression models, the utilization status of each input factor can be obtained, and the fishing output under 100% utilization rate of the input factor can be predicted. It provides help, reference for the next step of input factor control, and catch limit.

### 4.4. Support for the Management of the China Fishery

Estimation of harvesting capacity in fisheries is becoming increasingly important as nations address the problems associated with overharvesting. Acceptable and practical methods for estimating capacity have to be developed [61]. In this paper, we demonstrated how the utilization status of each input factor could be estimated using the DEA.

Since the "13th Five-Year Plan" period, based on improving fishery law and adhering to the fishing licensing system, China's management policy for offshore fishery has mainly been placed in input-based management mode. Major policies were implemented, including implementing total management of marine fishery resources in 2017 and implementing quota management for the offshore annual catch. A "dual control" policy was implemented for offshore fishing vessels, which means that the number and power of offshore fishing vessels are controlled, and the number and power of fishing vessels were reduced within five years [62–64]. These two policies explicitly put forward for the first time intended to reduce the amount and power of offshore fishing vessels with a specific number and offshore annual quota in the history of Chinese fishery management [65,66].

With "dual control" and end-of-year annual catch limitations, it also includes the promotion of the standardization of mesh size, the implementation of fishing ban, pilot fishing quota, and other corresponding policies and implementations [67–69]. In the next five years period, the "14th Five-year Plan" includes the management of China's offshore fishery aims with respect to controlling fishing capacity and strengthening conservation and sustainable utilization of resources. Therefore, according to the above-mentioned management policies and the combined with the results of this paper, suggestions and

support can be provided for the management of offshore fishery during the "14th Five-Year Plan" period.

First, the problem is whether or not to continue to reduce the number and power of offshore fishing vessels in the "dual controls" policy. According to the calculation results of Table 3 and Figure 3, the capacity utilization had an annual average of 93.5%. The idle rate of fishing vessels is one of the important indicators for the EU in inspecting overfishing in member states [70]. During the period of the "13th Five-Year Plan", the total number, the total tonnage, total power of fishing vessels, and the number of fishermen were all in excess. Therefore, during the "14th Five-Year Plan" period, we propose some suggestions with respect to continuing to adhere to the "dual control" policy: 1. The number of fishing vessels and power input should be continually reduced. 2. The renovation project of offshore fishing vessels should continue to be implemented. 3. Reconstruction of environmentally friendly resource-saving fishing vessels should be carried out. 4. Further improvements in safety, comfort, energy saving, and environmental protection characteristics of ocean fishing vessels should be performed. 5. Pelagic fishery production should be increased. 6. Fishermen's landing and transfer of businesses, etc., should improve.

Secondly, there is a problem with respect to the quota of offshore fishing at the end of the five-year plan. According to formula (8), a reduction in the above input factors can cause the annual catch to decrease continuously and achieve the target of catch limitation at the end of the "five-year period." For example, the number of fishing vessels should be further reduced by 20,000, the total power of fishing vessels should be further reduced by 1.5 million kilowatts, and the number of fishing fishermen should be reduced by 200,000 in the next five-year plan period. Therefore, total tonnage needs will be reduced to 4,999,077, and this will reduce 3 million tonnage from the current base. At this time, the annual catch can reach 9,005,000. In the next five-year plan, the number of ships will be reduced by 20,000, the power will be at 1.5 million kw, the total ton will be 3 million, the number of fishermen will be 200,000, and the final catch amount will reach up to 9 million tons. According to the implementation during the "13th Five-Year Plan" period, if the quantitative method continues to be adopted, the reduction in gross tonnage factor of fishing vessels will a difficulty and focus of the implementation. Therefore, more effective management has to be considered.

Thirdly, there is a problem related to the improvement and optimization of "dual control" policy. At present, "dual controls" refers to controlling the number and power of fishing vessels. Beneficial results have been achieved over the years, but there are still some limitations. For example, the gross tonnage of fishing vessels has increased significantly, and it is difficult to control the quantity of gross tonnage and its problems such as large machines with error labels still existing. Therefore, according to the results of this paper, DEA calculation shows that the factors affecting offshore fishing capacity include number, power, and total ton of fishing vessels. Therefore, considering the next "dual control" decree, we suggest adding the gross tonnage of fishing vessels based on the number and power of fishing vessels.

Finally, the combination of input and output management in China's Offshore Fishery should strengthen. From the perspective of fishery management policies of developed countries in the world, such as Europe and the United States, the balance between input management and output management is achieved according to the characteristics of their own fisheries. Most developed countries tend to adopt the mode of output management, such as quota fishing, as the main mode, and input management is adopted as the supplementary mode, which has achieved beneficial effects [1,71]. In this study, it has been demonstrated that it is difficult to continually push forward the "dual control" idea and a quantitative reduction in the number, power, gross tonnage, and final catch of fishing vessels. Therefore, based on the experience of the pilot study on quota fishing, we suggest strengthening output management in appropriate areas, including policies such as catch limits and catch monitoring, and facilitating programs that facilitate production-based management. For example, the limited fishing area should be encouraged and supported,

and the management of catch should be improved to promote effective supervision of catch, which needs continuous coordination with respect to balance and coordination between input-based management and output-based management of China's offshore marine fishery industry. Moreover, exerting joint efforts to realize the sustainable utilization of offshore fishery resources will be necessary.

## 5. Conclusions

The importance and practical value of offshore fishing capacity analysis for sustainable use and conservation of marine fishery resources cannot be overemphasized. DEA was used to measure fishing capacity based on the 2004–2020 *China Fishery Statistical Yearbook* and by using the number of fishing vessels, total power, total tonnage, and the number of professional fishermen as input measures. Moreover, annual catch was used as the output measure. Capacity utilization was calculated in a range from 80.7 to 100%, and its average is 93.5%. The utilization status of each input index was obtained, especially the large surplus of the total ton and power of fishing vessels, in order to investigate the advantages and disadvantages of the management measures of China's offshore marine fishing industry in the past and present and to examine the key problems that have to solved in the next step. Four regression models were compared, including Lasso, Ridge, KNN, and Polynomial Features. The polynomial model shows a higher goodness of fit and has more advantages in forecasting annual catches. Furthermore, the combined application of DEA and Polynomial model was used to analyze and discuss the management policies of China's offshore fishery, which can provide help and reference for future management. The final situation shows that gross tonnage and power of fishing boats have a large surplus, and the rising trend of gross tonnage is greater than the total power. Obviously, "dual control" and quantitative reduction in the number, power, gross tonnage, and final catch of fishing vessels are difficult to achieve. Therefore, we recommend strengthening production management in appropriate areas, including policies such as fishing restriction and fishing monitoring, and promoting production-based management. The next step will focus on the use of statistical data, catch data, and other multi-source data; using artificial intelligence and machine learning algorithms; and conducting in-depth research on the management and control of the fishing capacity of marine fishing vessels.

**Author Contributions:** Conceptualization, S.L., C.L. and T.-J.C.; methodology, S.L., C.L. and J.-X.S.; software, J.-X.S.; validation, S.L. and C.L.; formal analysis, S.L., C.L. and T.-J.C.; resources, S.L., J.-X.S., C.L. and H.-X.Z.; data curation, S.L., J.-X.S., C.L., and H.-X.Z.; writing—original draft preparation, S.L., C.L., T.-J.C. and H.-X.Z.; writing—review and editing, S.L., C.L., T.-J.C. and H.-X.Z.; visualization, S.L., C.L., T.-J.C. and H.-X.Z.; supervision, C.L. and T.-J.C.; project administration, S.L. and C.L.; funding acquisition, S.L. and C.L. All authors have read and agreed to the published version of the manuscript.

**Funding:** This work was supported by the Fishery administration Project (D8021210076) and Jimei University grant number C619061. The funders had no role in study design, data collection and analysis, the decision to publish, or the preparation of the manuscript.

**Data Availability Statement:** The data come from the data on offshore fishing operations in China obtained from the China Fishery Statistical Yearbook published from 2004 to 2020. China Fishery Statistical Yearbook is compiled by Fishery administration Bureau of Ministry of Agriculture and Rural Affairs, National Fisheries Technology Extension Center and Chinese Fisheries Association, China Agriculture Press, Beijing, China.

**Acknowledgments:** We thank Li Yu-wei for his contributions to the suggested revision comments to manuscript. Useful suggestions from anonymous reviewers were incorporated into the manuscript.

**Conflicts of Interest:** The authors declare no conflict of interest.

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
