# Peer review of "Evaluating Fishing Capacity Based on DEA and Regression Analysis of China’s Offshore Fishery"

_jmse, doi:10.3390/jmse9121402_

Round 1

Reviewer 1 Report

Thank you for providing your manuscript to the Journal of Marine Sceinces
and Engineering. There are fundamental issues with that manuscript. One
main issue is that it is totally focused on China, and not to Taiwan as
the Special Issue title indicates. Moreover, also the content deals only
with Chinese subjects without any comparison with the rest of the world.
This is also one of the reasons why the manuscript does not really have
a relevance for readers from the rest of the world. I strongly propose
to substantially improve the manuscript through a comprehensive comparison with other regions in the world. Moreover, the manuscript
does not respect Scientific Best Practice as the conclusions are missing. Furthermore, there are more serious issues which do not allow the manuscript in the current version to be published. The main issue in this regard is the extremely poor literature review. In a literature review, it is important that the scientific novelty of the work is established through a critical analysis of the related literature. If the authors would have done this, they would have to recognise that the manuscript does not have the quality for the classification as "article" (like given above the title) but a case report. There have been already done a lot of comparable studies around the world. Thus,
the main questions of the reviewer are: Which scientific question shall
be answered with the manuscript? What is your scientific hypothesis
that you wish to answer with the investigation? What is the novelty
that distinguishes this study from the number of comparable studies?
Furthermore, serious scientific sources from outside China must be
added to arrive to a comprehensive literature research. The methodology must be substantially improved. In the current form the methodology remains unclear. I strongly recommend to include a flow chart illustrating the steps of the methodology.

Reviewer 2 Report

The manuscript needs to be revised and shortened. There is a lot of repetition and in many cases the sections are not properly included where they belong. In particular, the results include material that belongs in the introduction or repeat information already presented in the introduction. The discussion includes results and really only begins with section 4.4 on line 516 in the attached. Following revision, the manuscript should be reviewed again to see if it is ready for publication. I found the approach interesting and see value in publishing but reject the manuscript in its current form.

Round 2

Reviewer 1 Report

Thank you for providing the revised vesion. The manuscript was substantially improved.. My comments have been addressed.